# Processing Conditions of a Medical Grade Poly(Methyl Methacrylate) with the Arburg Plastic Freeforming Additive Manufacturing Process

**DOI:** 10.3390/polym12112677

**Published:** 2020-11-12

**Authors:** Lukas Hentschel, Frank Kynast, Sandra Petersmann, Clemens Holzer, Joamin Gonzalez-Gutierrez

**Affiliations:** 1Polymer Processing, Montanuniversitaet Leoben, Leoben 8700, Austria; clemens.holzer@unileoben.ac.at; 2Additive Manufacturing, ARBURG GmbH + Co KG, 72290 Lossburg, Germany; frank_kynast@arburg.com; 3Materials Science and Testing of Polymers, Montanuniversitaet Leoben, Leoben 8700, Austria; sandra.petersmann@unileoben.ac.at

**Keywords:** additive manufacturing, melt deposition, medical applications, poly(methyl methacrylate)

## Abstract

The Arburg Plastic Freeforming process (APF) is a unique additive manufacturing material jetting method. In APF, a thermoplastic material is supplied as pellets, melted and selectively deposited as droplets, enabling the use of commercial materials in their original shape instead of filaments. The medical industry could significantly benefit from the use of additive manufacturing for the onsite fabrication of customized medical aids and therapeutic devices in a fast and economical way. In the medical field, the utilized materials need to be certified for such applications and cannot be altered in any way to make them printable, because modifications annul the certification. Therefore, it is necessary to modify the processing conditions rather than the materials for successful printing. In this research, a medical-grade poly(methyl methacrylate) was analyzed. The deposition parameters were kept constant, while the drop aspect ratio, discharge rate, melt temperatures, and build chamber temperature were varied to obtain specimens with different geometrical accuracy. Once satisfactory geometrical accuracy was obtained, tensile properties of specimens printed individually or in batches of five were tested in two different orientations. It was found that parts printed individually with an XY orientation showed the highest tensile properties; however, there is still room for improvement by optimizing the processing conditions to maximize the mechanical strength of printed specimens.

## 1. Introduction

Additive manufacturing (AM), colloquially known as 3D printing, has started to be a useful tool for the production of medical devices. AM can allow patient-specific medicine and the manufacturing of medical devices with intricate design, which by other methods would be too costly to manufacture. Furthermore, AM technology can be used to shape 3D objects and, using unique materials, reversible-stimuli-responsive functionality can be achieved (i.e., 4D printing) [1]. However, there are still technical limitations of AM that need to be overcome to ensure a safe application in the medical field, such as the identification of materials that can be safely used in long term applications. Further essential aspects, such as identifying cleaning and sterilization methods that can be used with the different materials, have to be defined [2]. Another limitation is that materials need to be certified to produce parts for medical applications. This certification process is time-consuming and expensive. Therefore, material producers go through this certification process for very few materials, particularly for materials for which the certification costs can be recovered in a reasonable amount of time. Altering the material in any way and even altering the processing conditions can render the certification invalid. Thus, it would be reasonable to use materials that have already been certified for medical applications in general. That means that the additive manufacturing equipment should be an open platform, where any material can be used. One example of an open system AM process is the Arburg Plastic Freeforming (APF) process developed by Arburg GmbH +Co KG (Lossburg, Germany).

The APF is a novel material jetting additive manufacturing technology that enables the production of complex thermoplastic components using standard material pellets [3]. In terms of medical application, the processing of granules offers a considerable advantage since no further filament making is necessary. Thus, no further manufacturing step has to be certified for medical purposes. A plastification unit similar to that of an injection molding machine provides the molten material and the pressure for the deposition process. After plastification, the polymeric material enters the discharge unit, consisting of a nozzle and a piezo-electrical value, which opens the nozzle up to 250 Hz [4,5]. Since the nozzle opens and closes at such high frequencies, the extruded melt forms droplets instead of a continuous string as is the case of other melt deposition technologies such as material extrusion additive manufacturing (MEX) with filaments (also known as fused filament fabrication (FFF)). Hence, the APF process is sometimes referred to as droplet deposition modelling [6]. The produced droplets are positioned on a Cartesian moveable building platform to form a three-dimensional structure layer-by-layer. A schematic representation of the process is shown in Figure 1. One significant advantage of the APF systems compared to FFF is the higher density that can be achieved in the fabricated specimens. This higher density can lead to better mechanical performance [3,7].

The APF technology is an open system meaning that variations of almost all process parameters are allowed by the manufacturer. This flexibility results in a great variety of processable materials, but it also means that the processing parameters have to be identified by the user. However, some support in finding the right parameters are given directly from ARBURG. The influences and interactions between processing parameters and component quality must be analyzed to optimize the processing parameters and be able to process new materials successfully. Process parameter optimization is a complex task since these interactions are not yet well understood. Thus, the material qualification or optimization of the mechanical properties for new materials has to be done systematically, varying one parameter at a time [3]. Therefore, this investigation aims to give an example on the qualification process to obtain specimens with good dimensional accuracy made of a medical-grade thermoplastic, commonly used for injection molding and extrusion of medical diagnostic devices. Examples of the applications of the selected polymer include diagnostic test packs, microfluidics and crystallography trays [8]. The processing parameters obtained after this investigation provide a starting point to produce specimens with a good appearance, which do not necessarily have the best mechanical performance achievable by the APF process.

The selected material was a medical grade poly(methyl methacrylate) (PMMA) because PMMA is an inert, biocompatible, transparent, and robust thermoplastic [9]. Due to these properties, PMMA has been used in the medical field as intraocular and hard contact lenses, as bone cement for orthopedic and cranial implants, as an anchor in hip prostheses, in dental applications, or for vertebroplasties and kyphoplasties [10,11]. PMMA is one of the most widely used polymeric materials for the reconstruction of cranial defects, and its use in cranioplasty dates back to the 1940s [12]. The intraoperative fabrication of PMMA implants by hand is the most common method of manufacturing [13,14,15]. However, fabrication by hand is being replaced by the production of molds for casting PMMA implants via thermoforming [16], milling [17], wax elimination [18], and additive manufacturing methods [19,20,21,22,23,24,25]. Currently, the direct printing of implants is being investigated [26], and this investigation represents a preliminary study to find suitable grades of PMMA for implant applications.

## 2. Materials and Methods 

Pellets of an amorphous thermoplastic compound based on PMMA used in the medical diagnostic industry (CYROLITE^®^ MD H12, Roehm GmbH, Darmstadt, Germany) were selected for this investigation. It has to be noted that this grade of PMMA has not yet been approved for permanent (i.e., more than 30 days inside the body) implants, but rather for other medical devices. All CYROLITE^®^ materials have been approved for food contact, and they are USP Class VI and ISO 10993-1 certified [8]. Nevertheless, the use of this grade as a permanent implant has not been discarded. That is why this investigation and many others are being performed as part of the CAMed (Clinical additive manufacturing for medical applications) project [27] to analyze its suitability as an implant material. Some of the relevant physical properties as supplied by the manufacturer are shown in Table 1.

An Arburg freeformer 200-3X (ARBURG GmbH + Co KG, Lossburg, Germany) additive manufacturing machine was used to fabricate specimens. The print job was prepared in the Arburg freeformer software v2.30 (ARBURG GmbH + Co KG, Lossburg, Germany). The diameter of the nozzle used was 0.2 mm. 

The APF process has many process parameters that need to be adjusted for a successful fabrication process. These processing parameters are specific for a given polymer type and even for a given grade of material since they depend on the surface tension, viscosity, melt strength, and thermal properties of each material. For this investigation, the initial printing conditions for another non-medical type of PMMA previously tested by ARBURG were used. These processing conditions are shown in the second column of Table 2.

A brief description of the APF processing parameters is given here. The position of the different heated zones (T_1_, T_2_, T_nozzle_ and T_chamber_) is shown schematically in Figure 1. Dosing stroke, similar to injection molding, is the distance that the screw travels backwards and controls the volume of the shot. The backpressure is the applied pressure to keep the screw secure during the deposition process. The screw speed defines the turning speed of the screw and is equal to the speed occurring on the bottom of the flow channel. The discharge rate is the volume of material being deposited. The drop overlap is how much the drops overlap each other during deposition. The drop aspect ratio (DAR) is the ratio between the width (B) and the height (H) of the droplet being extruded from the nozzle of the APF (insert in Figure 1 and Figure 2). The DAR is influenced by the material properties and the processing conditions. The layer height is the distance the build platform moves down during the deposition to determine the layer thickness. 

Using the initial processing values shown in Table 2 and with the help of an optical microscope (Figure 2), the initial drop aspect ratio (DAR) was estimated to be 1.26. 

Since it was observed that the initial values shown in Table 2 did not yield accurate specimens for the medical grade PMMA, the DAR, the discharge rate, melting temperature, build chamber temperature, and droplet overlap were adjusted systematically. After each adjustment was made, the height along the x- and y-direction of each printed specimen was measured at 25 spots using an analogue dial gauge with a measuring range between 0.01 and 10 mm (No. 2048-10, Mitutoyo Corporation, Kanagawa, Japan). 

Cube specimens with dimensions 20 mm × 20 mm × 20 mm were printed with conditions between the initial and final values in Table 2 to check the geometrical accuracy. Additionally, dog bone specimens, according to ISO 527-2 1A, were printed with the “PMMA final values” in Table 2 to characterize the tensile properties of the specimens with excellent dimensional accuracy. All specimens were built up with a single contour line and a 100% and ±45° rectilinear infill strategy. Two building orientations were investigated and labelled according to the plane of the silhouette, thus, XY was used for the laying samples and XZ for the standing samples on the long edge (Figure 3a). Furthermore, single and multiple parts were fabricated at the same time to study the influence of the batch size. In total, 60 dog bone specimens were printed and distributed in different batches, as shown in Table 3. The water-soluble support material ARMAT11 (ARBURG GmbH + Co KG, Lossburg, Germany) was used beneath the parallel zone of the tensile specimen and on the grip zone to prevent tilting and platform detachment of the specimens during printing (yellowish material in Figure 3b) to print specimens in the XZ orientation successfully. The values of the processing parameters for the support material are shown in Table 2.

Tensile testing was performed on the universal testing machine Zwick Z250 (ZwickRoell GmbH + Co KG, Ulm, Germany) at a testing speed of 1 mm/min until an elongation of 0.25% was reached for measuring of the tensile modulus and 50 mm/min afterwards until rupture occurred. The use of two testing speeds is in accordance with ISO 527-1, and it is used to speed up the testing procedure. The maximum loading with this machine is limited to 10 kN. Mechanical grips were used for clamping, and the deformations were evaluated by digital image correlation using a Mercury RT System (Sobriety s.r.o., Kuřim, Czech Republic).

## 3. Results

### 3.1. Optimized Processing Parameters

The printing results using the “PMMA initial values” according to Table 2 that had a discharge rate of 70%, yielded a cube that seems to be overfilled and shows warped corners (upper row in Figure 4). Besides, the print job stopped with warning errors such as “discharge value out of range” or “axis reading errors”. Since this was unacceptable, the discharge rate was reduced from 70 to 65%. This discharge reduction led to a more stable process with no errors or warped corners (lower row in Figure 4).

After further inspection of the cubes, it was observed that the drops did not weld together sufficiently. This insufficient welding had to be improved by increasing the energy input. Thus, the temperature of the building chamber was increased to 120 °C. Increasing the build chamber temperature led to a better appearance, but the welding was still not good enough. Therefore, the temperature of zone 1 was increased from 195 to 200 °C and the temperature of zone 2 from 225 to 230 °C. Also, the nozzle temperature was increased from 240 to 245 °C to decrease the process pressure. The cubes produced with the increased temperatures still showed warping and tapered edges regardless of the DAR used. Thus, the new strategy was to decrease the building chamber temperature from 120 to 100 °C, and the drop overlap from 50 to 25%. The resulting print jobs for all chamber temperatures and DARs are given in Figure 5.

The height at 25 locations along the top surface was measured to get a more accurate picture of the geometrical accuracy of the printed cubes. Figure 6 shows the cubes printed with different DAR (1.25 to 1.30) and T_chamber_ of 120, 110 and 100 °C. Please note that the surface roughness on the specimens was not considered in the height measurements since the roughness is smaller than the resolution of the measuring device used. As can be seen in Figure 6, reducing the chamber temperature led to a more uniform height. Variations in the DAR were also necessary to improve the flatness of the top surface. However, the DAR that improves the flatness was also dependent on the chamber temperature. For example, the most uneven surface for cubes printed at a chamber temperature of 120 °C was reached with a DAR of 1.27. On the other hand, at a chamber temperature of 110 °C, the most uneven surface was measured when the DAR was 1.25. Finally, when the chamber temperature was 100 °C, the most uneven surface was obtained with a DAR of 1.26. Therefore, there is no clear trend regarding DAR and surface evenness. Nonetheless, it was observed that a cooler chamber resulted in improved evenness and thus geometrical accuracy. However, care should be taken not to go too low as there could be low adhesion between the droplets and layers.

As a final step, the discharge rate was varied from 65 to 68% in increments of 1% to obtain better adhesion between droplets and layers. Figure 7 shows the resulting cubes at three different discharge values (65%, 66% and 67%). It has to be mentioned that a discharge value of 68% was not possible due to an axis error during the build job. Thus, 67% was set as the discharge value. The specimen processed with DAR = 1.29 had the best appearance with an even color and the best evenness.

Since thermoplastics shrink after fabrication, this shrinkage has to be compensated by using a scaling factor in the slicing software. For this purpose, five cubes 30 mm × 30 mm × 30 mm were printed individually using the final values shown in Table 2. The dimensions of the specimens were measured in the three axes (Figure 8). It was found that there was anisotropic shrinkage with the cube shrinking more in X- and Y-direction. Therefore, scale factors for each of the directions were calculated: X-direction = 1.015, Y-direction = 1.015, and Z-direction = 1.00 to improve the geometrical accuracy of the printed specimens.

Tensile specimens and other geometries such as possible cranial or dental implants were printed using the optimized values of the processing parameters (i.e., “PMMA final values” in Table 2). Examples of such printed specimens are shown in Figure 9. The quality of the printed geometries was considered acceptable based on their appearance, but further characterization on the parts to determine their mechanical performance, internal microstructure, bio-compatibility, and long term stability is ongoing and will be reported in future publications.

### 3.2. Tensile Properties

All tensile specimens were printed with the final processing values in Table 2, which were determined to obtain good geometrical accuracy. Printed dog bone specimens with different building orientations (XY or XZ) and produced in batches of one (Single) or five (Multi) specimens were tested. Examples of the stress-strain curves and the calculated tensile modulus, tensile strength and tensile elongation at break for specimens printed with different parameters are shown in Figure 10. Please notice that the exact conditions (e.g., strain rate) at which the producer tested their specimens are not known, and since the tensile properties of PMMA are rate dependent, this might have an effect on the actual values of the specification. Therefore, the values provided are illustrative only to indicate that there is room for improvement that can be archived by modifying the printing parameters.

In Figure 10, it can be seen that the orientation of the parts affects the measured tensile properties. For example, the tensile values for the specimens printed in the XY-plane appear to be slightly higher than those printed in the XZ-plane. However, the only significantly different values are the tensile strength (Figure 10c) and the tensile elongation at break (Figure 10d) for the specimens printed individually. These results suggest that the specimens printed in the XY-orientation are exposed to a more intense localized heating that promotes a better droplet welding, which leads to higher strength and strain at break. A denser structure can be observed mainly in the surface of the parallel zone of the tensile specimens, where the specimens have sections with higher transparency when printed in the XY-orientation due to better welding (Figure 11a,b), and more opaque sections when printed in the XZ-orientation, since the individual layers are visible and diffract the light (Figure 11c,d). Specimens printed in the XZ-orientation have a larger surface area to volume ratio exposed to the surrounding air in the build chamber, resembling the fins of a convection heat exchanger. Therefore, even though the path within one layer is smaller, the welding to the next layer is weaker since the temperature of the previous layer might be colder, which leads to weaker tensile properties.

The effect of the build orientation has already been investigated for ABS in the APF process [3]. It was observed that samples printed in the XY-orientation showed considerably higher values for the tensile strength, elongation at break and the tensile modulus compared to parts built in YZ-orientation (i.e., upright orientation). Similar results were observed in studies dealing with FFF of polylactide (PLA) [28] and amorphous polyetherimide (PEI) [29]. For this reason, the YZ-orientation was not considered in this study. Additionally, the FFF studies [28,29] compared the mechanical properties of printed parts in the XY-plane and the XZ-plane, and it was observed that the XZ-orientation showed the highest mechanical properties. It is essential to mention that for PLA specimens, the strands were deposited in the loading direction (at 0°), while for PEI specimens, the strands were deposited at 0° or ±45°. Regardless of the deposition angle, the specimens printed in the XZ-plane were stronger than the specimens printed in the XY-plane [29]. However, based on the work done using ABS in the APF [3], it appears that the angle of deposition that yields the highest tensile properties is 90°, which is the complete opposite to the one that yields the maximum tensile properties in FFF (i.e., 0°). Therefore, it can be expected that the results obtained with parts produced by APF do not necessarily follow the same trends as the parts made by FFF since the deposited shapes are not the same (i.e., strings of droplets for the APF and cylindrical rods for FFF) and the infill orientations are not equal. Another difference between the FFF printers used in the two studies and the APF is that the build chambers are of different dimensions and for the printers used to produce PLA specimens, the temperature was not actively controlled. These discrepancies make a direct comparison between the different studies difficult. 

In general, the “Single” printed parts show slightly higher properties compared to the parts produced in “Multi”-part batches (Figure 10). However, the results are not significantly different. The only statistically different results are the tensile strength and elongation at break between the single printed specimens in the XY-plane and XZ plane (Figure 10c,d). The different thermal history can explain the variability in results. During single part print jobs, the thermal history for each specimen is more similar to each other. Therefore, each specimen produced in separate batches is more similar to each other, and thus the standard deviation for the tensile properties is smaller. In contrast, the thermal history of each specimen within a multi-part batch is slightly different at each position on the build platform. These different thermal histories lead to higher variability in the mechanical data. Furthermore, the individually printed parts have better welding between the different layers since it takes less time to build a new layer on top of the previous one. That way, the previous layer is colder, and the contact temperature is lower [30] when printing multiple parts batches than when printing one specimen at a time. This difference in welding can be observed in the specimens shown in Figure 12. The specimens printed individually are more transparent (Figure 12a,b) than the ones printed in batches of five specimens (Figure 12c,d). This difference in transparency suggests better welding between droplets and layers for single printed specimens.

When comparing the measured tensile properties to the specifications supplied with the raw material, it was found that parts printed individually with an XY-orientation showed the highest tensile properties with a tensile modulus of 93% and tensile strength of 72% in comparison to the supplier’s specifications (Table 1). As with many other additively manufactured specimens, the elongation at break is low for the APF specimens, having a strain at break of only 42% of that given in the specification for the raw PMMA material. Thus, it is clear that similar to FFF [31], the processing conditions that lead to the best appearance do not necessarily lead to the optimal tensile properties. There is always a trade-off between geometrical accuracy and mechanical properties in AM technologies since the shape cannot be constrained as it happens inside a molding tool. Therefore, further research is needed to optimize the tensile performance of medical-grade PMMA in APF. 

## 4. Conclusions

This investigation illustrates a possible way to qualify medical grade materials for the Arburg Plastic Freeforming (APF) process and obtain geometrically accurate specimens. The research also shows the versatility of the APF process to use thermoplastic materials that have not been specifically tailored for an additive manufacturing process in their standard granular form. The material selected to illustrate the qualification process was a medical-grade poly(methyl methacrylate) (PMMA) material used in the injection molding or extrusion of medical diagnostic devices. The qualification processes started by estimating an initial drop aspect ratio (DAR) by extruding a string of droplets at processing conditions for another non-medical grade PMMA. Printing with the initial conditions led to overfilled specimens with warped corners, thus the discharge rate was decreased, the processing temperatures were increased, and the DAR was varied at different temperatures. The chamber temperature was varied, and the DAR was readjusted until the printed specimens had a smooth surface. Finally, the anisotropic shrinkage was compensated in each printing axis to improve the geometrical accuracy. Once the processing conditions were defined, exemplary geometries for orthopedic implants and tensile specimens were printed in different orientations and batches with a different number of specimens. The highest tensile properties were obtained for parts printed in batches consisting of one specimen in the XY-printing orientation (i.e., laying on the platform). Since the used processing conditions were selected to give specimens with good geometrical accuracy, the tensile properties were not maximized. Therefore, a future investigation of the processing parameters is planned to maximize the tensile properties. 

## Figures and Tables

**Figure 1 polymers-12-02677-f001:**
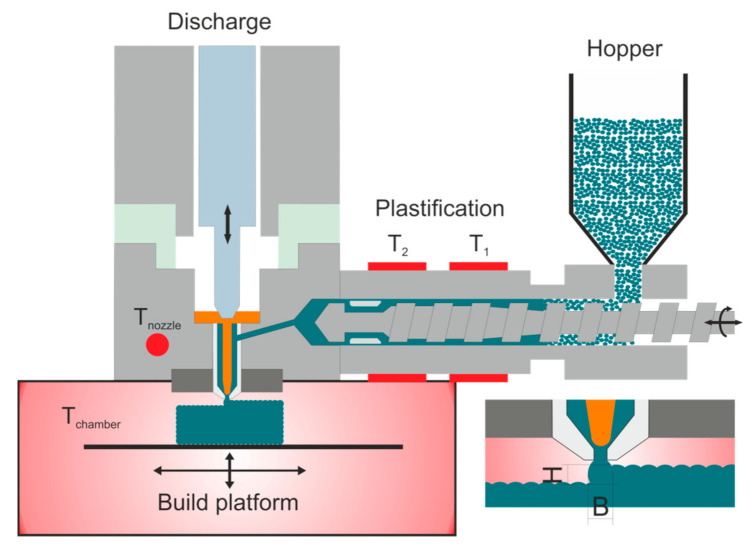
Working principle of Arburg Plastic Freeforming process where T_1_ is the temperature in cylinder zone 1, T_2_ is the temperature in cylinder zone 2, T_nozzle_ is the nozzle temperature, and T_chamber_ is the chamber temperature. B and H are the width and height of the produced droplets, respectively.

**Figure 2 polymers-12-02677-f002:**
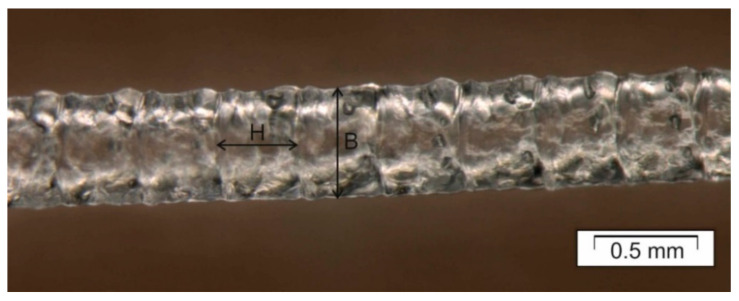
Micrograph of a string of droplets obtained using the initial printing conditions in Table 2. H values are from 0.22 to 0.25 mm and B values from 0.28 to 0.32 mm, giving a drop aspect ratio (DAR = B/H) of approximately 1.26.

**Figure 3 polymers-12-02677-f003:**
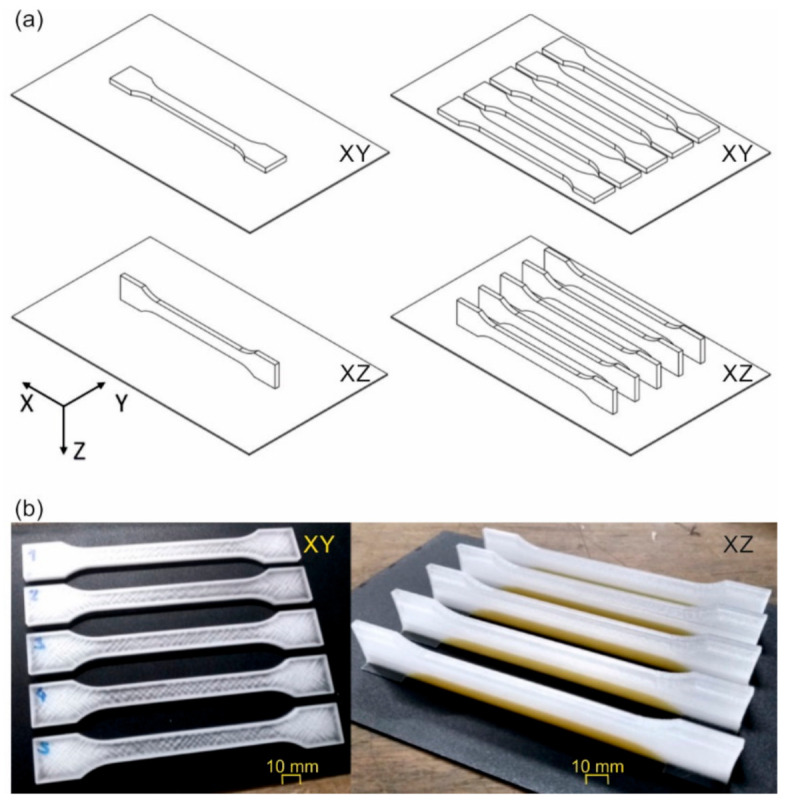
(**a**) Schematic representation of different orientations (XY or XZ) and arrangements (single or multiple) of tensile bars during printing, and (**b**) examples of printed parts without support material and with support material (yellowish material).

**Figure 4 polymers-12-02677-f004:**
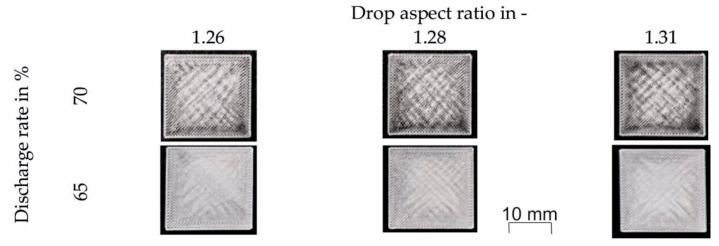
Cubic specimens printed at a chamber temperature of 100 °C with three different drop aspect ratios (1.26, 1.28, and 1.31) and discharge rates of 70 and 65%. Other processing temperatures were T_1_ = 195 °C, T_2_ = 225 °C, and T_nozzle_ = 240 °C.

**Figure 5 polymers-12-02677-f005:**
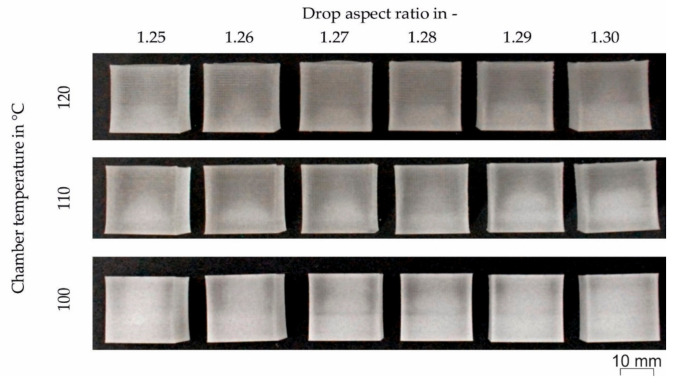
Printed cubes with six different values for the drop aspect ratio between 1.25 and 1.30, and three different chamber temperatures (T_chamber_) and decreased drop overlap of 25%. Other processing temperatures were T_1_ = 200 °C, T_2_ = 230 °C, and T_nozzle_ = 245 °C.

**Figure 6 polymers-12-02677-f006:**
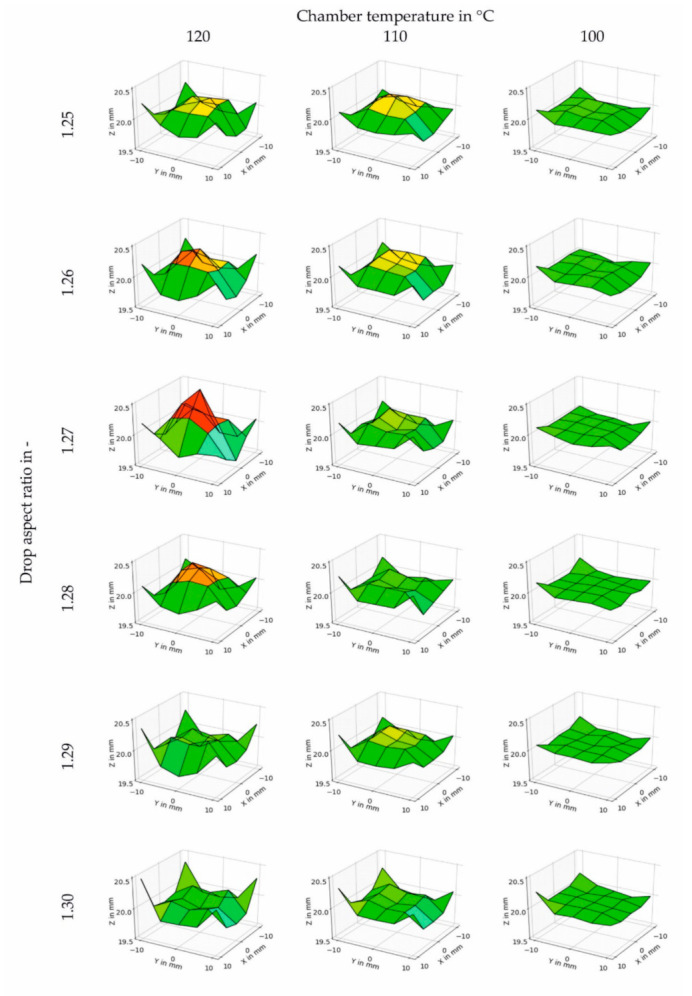
Surface plots of the height measured for cubic specimens printed at different chamber temperatures between 120 and 100 °C and different drop aspect ratios between 1.25 and 1.30. Other processing temperatures were T_1_ = 200 °C, T_2_ = 230 °C, and T_nozzle_ = 245 °C.

**Figure 7 polymers-12-02677-f007:**
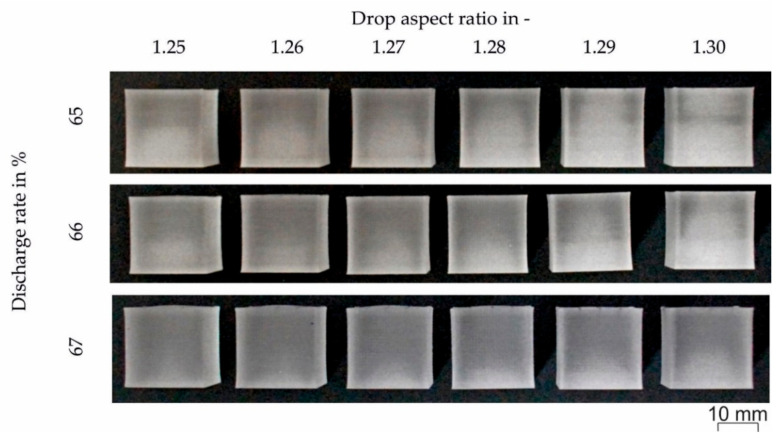
Cubic specimens printed at different discharge rates (65, 66 and 67%) and drop aspect ratios between 1.25 and 1.30. Other processing temperatures were T_1_ = 200 °C, T_2_ = 230 °C, T_nozzle_ = 245 °C, and T_chamber_ = 100 °C.

**Figure 8 polymers-12-02677-f008:**
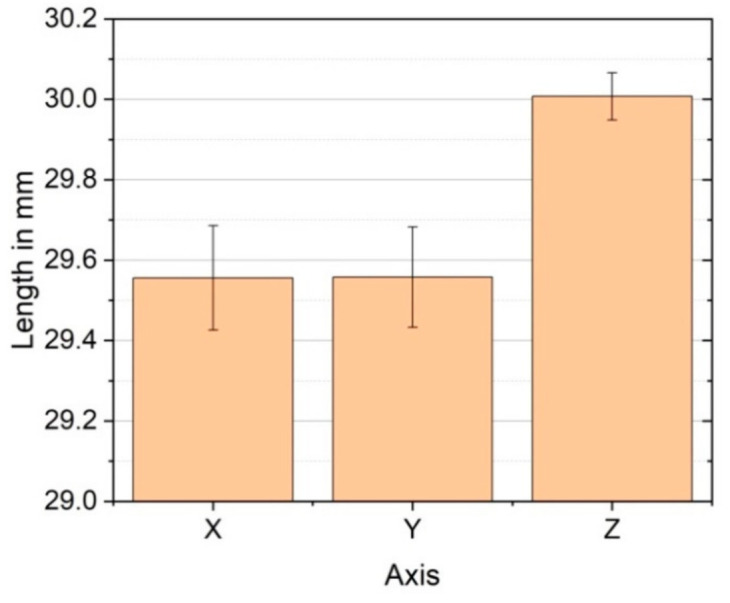
Shrinkage in each printing direction for cubic specimens 30 mm × 30 mm × 30 mm.

**Figure 9 polymers-12-02677-f009:**
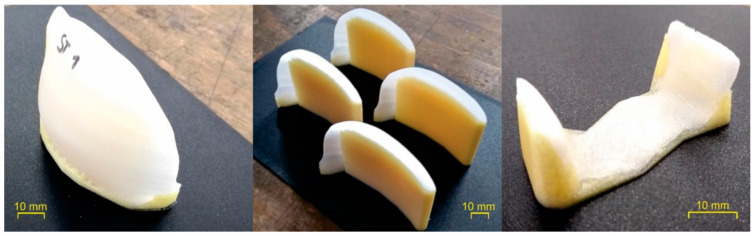
Benchmark “implants” printed with the optimized printing parameters (“PMMA final values” in Table 2).

**Figure 10 polymers-12-02677-f010:**
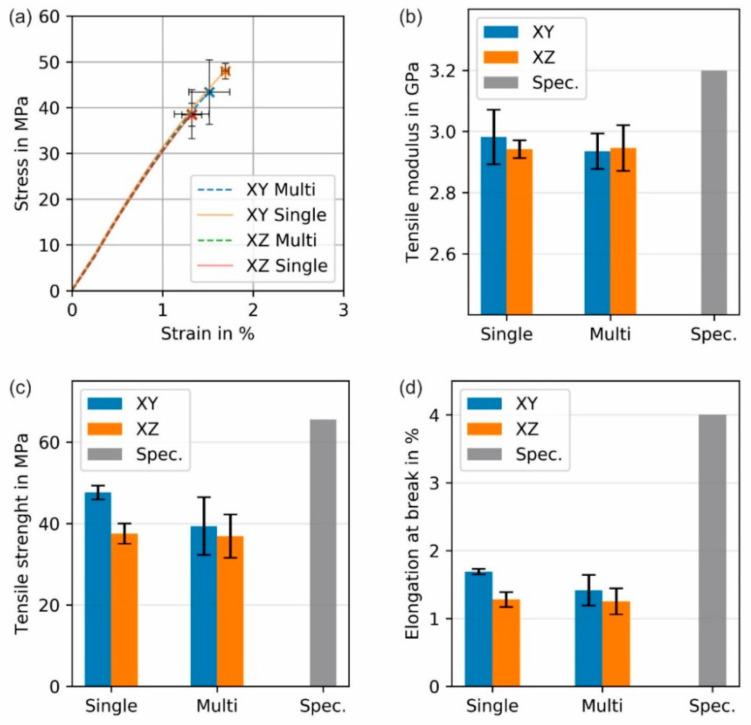
Tensile mechanical properties of printed specimens in two orientations (XY and XZ) and the batch of one specimen (Single) or five specimens (Multi): (**a**) stress-strain curves, (**b**) tensile modulus, (**c**) tensile strength, and (**d**) elongation at break. Specifications (Spec.) for the neat (most likely injection molded PMMA) given by the producer are given for comparison. Individual stress-strain curves can be seen in the Appendix A.

**Figure 11 polymers-12-02677-f011:**
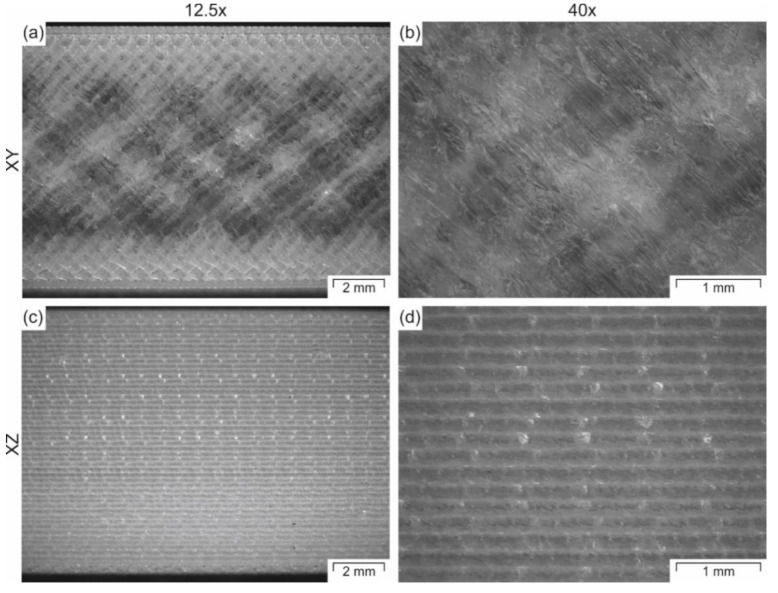
Microscopy images of tensile specimens printed in (**a**,**b**) XY-orientation or (**c**,**d**) XZ-orientation recorded at two different magnifications (12.5 and 40). All images were taken from the top view and laying the specimens in the XY-plane.

**Figure 12 polymers-12-02677-f012:**
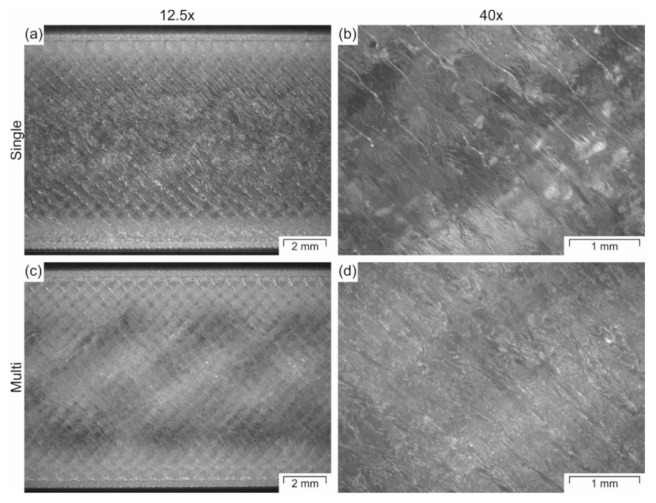
Microscopy images of tensile specimens printed in (**a**,**b**) batch of one specimen (Single) or (**c**,**d**) five specimens (Multi) recorded at two different magnifications (12.5 and 40). All images were taken from the top view and laying the specimens in the XY-plane.

**Table 1 polymers-12-02677-t001:** Tensile properties as supplied by the producer.

Parameter	Testing Method	Typical Value
Tensile Strength	ASTM D 638	65.5 MPa
Tensile Modulus	ASTM D 638	3.2 GPa
Tensile Elongation at Break	ASTM D 638	4–6%
Vicat Softening Point 1.8 MPa	ASTM D1525	105 °C
Melt Flow Rate 230 °C & 3.8 kg	ASTM D1238	7.0 g/10 min
Specific Gravity	ASTM D 792	1.19

**Table 2 polymers-12-02677-t002:** Printing conditions for PMMA and the support material.

Parameter	PMMA Initial Values	PMMA Final Values	Support Material Values
Temperature zone 1 (T_1_) in °C	195	200	140
Temperature zone 2 (T_2_) in °C	225	230	180
Nozzle temperature (T_nozzle_) in °C	240	245	200
Chamber temperature (T_chamber_) in °C	100	100	100
Dosing stroke in mm	8	8	6
Backpressure in bar	30	40	50
Screw speed in m/s	4	4	8
Discharge rate in %	70	67	100
Droplet overlap in %	50	25	40
Drop aspect ratio (DAR) in -	1.26	1.29	1.65
Layer height in mm	0.2	0.2	0.2
Scale factor X-direction in -	1.000	1.015	1.000
Scale factor Y-direction in -	1.000	1.015	1.000
Scale factor Z-direction in -	1.000	1.000	1.000

**Table 3 polymers-12-02677-t003:** The number of batches and specimens prepared in different orientations.

Build Orientation	* Processing Conditions	Total Single Batches	Total Single Specimens	Total Multiple Batches	Total Multiple Specimens
XY	PMMA final	15	15	3	15
XZ	PMMA final	15	15	3	15

* Processing conditions as defined in Table 2.

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
