# Peer review of "Processing Conditions of a Medical Grade Poly(Methyl Methacrylate) with the Arburg Plastic Freeforming Additive Manufacturing Process"

_polymers, 2020, doi:10.3390/polym12112677_

Round 1

Reviewer 1 Report

The manuscript talks about a current research topic and shows news in terms of research and development.
The results presented are supported by the methodology (which was written in detail).
The processing conditions were defined, tested and compared for the production of polymeric medical parts. The authors show the advances and limitations of the produced materials, with prospects for optimization.
The authors actually show how to qualify medical grade materials based on PMMA for the Arburg Plastic Freeforming (APF) process and obtain geometrically accurate specimens for potential applications.
I indicate the publication of the manuscript, as it was written.

Author Response

The manuscript talks about a current research topic and shows news in terms of research and development.
The results presented are supported by the methodology (which was written in detail).
The processing conditions were defined, tested and compared for the production of polymeric medical parts. The authors show the advances and limitations of the produced materials, with prospects for optimization.
The authors actually show how to qualify medical grade materials based on PMMA for the Arburg Plastic Freeforming (APF) process and obtain geometrically accurate specimens for potential applications.
I indicate the publication of the manuscript, as it was written.

  1. We thank the reviewer for the positive evaluation of our work.

Reviewer 2 Report

The manuscript deals with the assessment of processing parameters and condition for medical grade PMMA in freeforming additive process. It is a well written and scientifically sound manuscript. The following comments should be addressed before acceptance.

  1. Tensile stress sraing graphe should be represented propertly to see the different ones.
  2. I see there is a difference in mechanical properties from spec and experiments. Is there any other reason behind such a difference apartf from the printing conditons?

Author Response

The manuscript deals with the assessment of processing parameters and condition for medical grade PMMA in freeforming additive process. It is a well written and scientifically sound manuscript. The following comments should be addressed before acceptance.

  1. Tensile stress-strain graph should be represented properly to see the different ones.

Thanks for the comment. We have separated the stress-strain curves into individual diagrams for better visibility and added the curves as supplementary information (Figure S1) since the values that can be extracted from these stress-strain curves can be seen in the three bar-charts provided in Figure 10. The caption of Figure 10 now reads: “Individual stress-strain curves can be seen in the supplementary information (Figure S1).”

  1. I see there is a difference in mechanical properties from spec and experiments. Is there any other reason behind such a difference apart from the printing conditions?

The authors agree that there is a difference between the experimental and the specifications given by the producer. The material is recommended for profile extrusion or injection moulding, so the specifications are not measured on additively manufactured specimens; thus the morphology and density of the specimens are very different, and therefore, this is the main factor contributing to the differences. Other factors that might influence the results are the conditions used for performing the tensile testing, such as temperature, humidity and cross-head speed. Therefore, we have modified the text before discussing the tensile testing results as follows (line 260): “Please notice that the exact conditions (e.g. strain rate) at which the producer tested their specimens are not known and since the tensile properties of PMMA are rate dependent this might have an effect on the actual values of the specification. Therefore, the values provided are illustrative only to indicate that there is room for improvement that can be archived by modifying the printing parameters.”

Reviewer 3 Report

Recommendation: Reconsider after major revisions

Comments: In the article titled “Processing conditions of a medical grade poly(methyl methacrylate) with the Arburg Plastic Freeforming additive manufacturing process”, the authors modified the processing conditions such as discharge rate, melt temperatures, build chamber temperature and evaluated the geometry and tensile properties of the final prints. However, there are some problems that need to be solved or questions need to be answered before it can be published in Polymers.

  1. Scale bar is needed for Figure 3, Figure 4, Figure 5, and Figure 7.
  2. What do the authors mean “Tensile testing was performed on the universal testing machine at a testing speed of 1 mm/min for measuring of the modulus and 50 mm/min afterward.” in Line 163-165? Were two stretching speeds used?
  3. Based on the micrograph in Figure 2, the string does not have a smooth surface. How does the surface roughness affect the error of the height measurement in Figure 6?
  4. The authors listed the tensile properties as supplied by the producer in Table 1 and used that as a reference to compare the tensile results of the printed tensile bars in Figure 10. Were the samples also 3D printed using the same technique by the producer? Also, tensile properties of plastics such as modulus are rate dependent. Was the same stretching speed used for the tensile measurements?
  5. The authors showed the printing direction in Figure 11 but should also note the imaging direction e.g. the images were taken from the top view. The same to Figure 12.
  6. The authors claimed, “the only significantly different values are the tensile strength and the tensile elongation at break for the specimens printed individually.” in Line 253-255. This is true. So the printing direction affects the tensile properties of the printed parts. However, the authors also stated “It can be seen that both the orientation and the number of parts per batch affect the measured tensile properties…” in Line 251-252. This statement does not hold. Based on the result in Figure 10, the number of parts per batch does not seem to affect the tensile results. The same to the discussion in Line 290-293. The tensile properties of the single printed part and the multi-part batches are not statistically significant different.

Author Response

Comments: In the article titled “Processing conditions of a medical grade poly(methyl methacrylate) with the Arburg Plastic Freeforming additive manufacturing process”, the authors modified the processing conditions such as discharge rate, melt temperatures, build chamber temperature and evaluated the geometry and tensile properties of the final prints. However, there are some problems that need to be solved, or questions need to be answered before it can be published in Polymers.

  1. Scale bar is needed for Figure 3, Figure 4, Figure 5, and Figure 7.

Thank you for the observation, we agree that scale bars are needed in the figures; therefore, we have added the scale bars to Figure 3, Figure 4, Figure 5, and Figure 7, as suggested, as well as in Figure 9.

  1. What do the authors mean “Tensile testing was performed on the universal testing machine at a testing speed of 1 mm/min for measuring of the modulus and 50 mm/min afterwards.” in Line 163-165? Were two stretching speeds used?

Yes, two stretching speeds were used during the same tensile test. The tests were started slowly at 1 mm/min up to an elongation of 0.25% and then increase the speed to 50 mm/min until rupture occurred. The text has been revised for clarity, and now it reads (line 170): “Tensile testing was performed on the universal testing machine Zwick Z250 (ZwickRoell GmbH + Co KG, Ulm, Germany) at a testing speed of 1 mm/min until an elongation of 0.25 % was reached for measuring of the tensile modulus, and 50 mm/min afterwards until rupture occurred.”

  1. Based on the micrograph in Figure 2, the string does not have a smooth surface. How does the surface roughness affect the error of the height measurement in Figure 6?

We agree with the reviewer that the specimens do not have a smooth surface, but the surface roughness is beyond the resolution of the measuring device used to measure the height of the specimens; therefore the surface roughness was neglected to obtain Figure 6. The following note has been added to the text (line 206): “Please note that the surface roughness on the specimens was not considered in the height measurements since the roughness is smaller than the resolution of the measuring device used.”

  1. The authors listed the tensile properties as supplied by the producer in Table 1 and used that as a reference to compare the tensile results of the printed tensile bars in Figure 10. Were the samples also 3D printed using the same technique by the producer? Also, tensile properties of plastics such as modulus are rate dependent. Was the same stretching speed used for the tensile measurements?

The material is not generally used for additive manufacturing, but rather for profile extrusion and injection moulding; therefore, the specimens from the supplier were not produced by additive manufacturing. A common practice is to inject mould the specimens for tensile testing. Yes, polymers are rate dependent; therefore the text has been revised to read as follows (line 260): “Please notice that the exact conditions (e.g. strain rate) at which the producer tested their specimens are not known and since the tensile properties of PMMA are rate dependent this might have an effect on the actual values of the specification. Therefore, the values provided are illustrative only to indicate that there is room for improvement that can be archived by modifying the printing parameters.”

  1. The authors showed the printing direction in Figure 11 but should also note the imaging direction, e.g. the images were taken from the top view. The same to Figure 12.

We agree with the reviewer´s comment; the imaging direction has been mentioned in the caption of Figures 11 and 12. The captions now have the following note: “All images were taken from the top view and laying the specimens in the XY-plane.”

  1. The authors claimed, “the only significantly different values are the tensile strength and the tensile elongation at break for the specimens printed individually.” in Line 253-255. This is true. So the printing direction affects the tensile properties of the printed parts. However, the authors also stated “It can be seen that both the orientation and the number of parts per batch affect the measured tensile properties…” in Line 251-252. This statement does not hold. Based on the result in Figure 10, the number of parts per batch does not seem to affect the tensile results. The same to the discussion in Line 290-293. The tensile properties of the single printed part and the multi-part batches are not statistically significant different.

We agree with the reviewer´s observations. The text (line 268) has been modified as follows: “In Figure 10, it can be seen that the orientation of the parts affects the measured tensile properties. For example, the tensile values for the specimens printed in the XY-plane appear to be slightly higher than those printed in the XZ-plane.”

And the text (line 306) was modified to read: “The only statistically different results are the tensile strength and elongation at break between the single printed specimens in the XY-plane and XZ plane (Figure 10c and Figure 10d).”

Reviewer 4 Report

1) English language requires editing since, although the paper is well written, it still holds many flaws. English editing is recommended for improvement on this specific query.

2) The novelty of the manuscript is unclear and under major question. What is the main difference between the present paper and the 3D/4D printing method and research works on the freeforming additive manufacturing process published by Bodaghi’s group? It needs to be clarified in the introduction section.

3) Literature review needs to be updated. Some of new references on 3D/4D printing can be found below:

https://www.tandfonline.com/doi/abs/10.1080/17452759.2020.1795209

https://iopscience.iop.org/article/10.1088/1361-665X/ab0b6b/meta

4) It is needed to have a section focused on the modelling of 3D printed samples. Check the above-mentioned papers.

5) The advantage of the present method compared with other available methods should be demonstrated.

Conclusion:
In view of the above, the manuscript is not recommended for publication in its present form. The authors should modify the manuscript and address the above comments. In my opinion, a revised version including above mentioned points may be considered for publication after reconsideration.

Author Response

  • The English language requires editing since, although the paper is well written, it still holds many flaws. English editing is recommended for improvement on this specific query.

Thanks for the observation, we have proof-read the manuscript and corrected the language by using grammar correction software. The corrections are visible in track changes throughout the manuscript.

2) The novelty of the manuscript is unclear and under major question. What is the main difference between the present paper and the 3D/4D printing method and research works on the freeforming additive manufacturing process published by Bodaghi’s group? It needs to be clarified in the introduction section.

In this manuscript, we concentrate on a particular technology which is called the Arburg Plastic Freeforming, which is described in detail in the manuscript, since it is not a very common technology. Bodaghi´s group is dealing with 4D printing along with many other researchers in the world; for this reason, these works were not initially considered. Nevertheless, we have modified the text to mention the possibility of having additional functionality on 3D printed parts.

The text has been updated as follows (line 37): “Furthermore, AM technology can be used to shape 3D objects and using unique materials, reversible-stimuli-responsive functionality can be achieved (i.e. 4D printing) [1]”.

3) Literature review needs to be updated. Some of the new references on 3D/4D printing can be found below:

https://www.tandfonline.com/doi/abs/10.1080/17452759.2020.1795209

https://iopscience.iop.org/article/10.1088/1361-665X/ab0b6b/meta

The review paper of Zolfagharian et al. 2020 was cited after the statement mentioning 4D printing (line 39) since it provides an up-to-date general overview on 4D printing, the second paper is from last year and we do not have a subscription to IOP Science; therefore it could not be downloaded and cited.

4) It is needed to have a section focused on the modelling of 3D printed samples. Check the above-mentioned papers.

The main goal of the manuscript is to present an experimental methodology to produce specimens with good geometrical accuracy with the APF machine using commercially available medical grade pellets of PMMA. The work is purely experimental; therefore, a simulation section is out of the scope of the manuscript. Therefore, we have decided not to consider this suggestion; this is an investigation for future works that require some collaborative efforts since it is not our primary expertise.

5) The advantage of the present method compared with other available methods, should be demonstrated.

Thanks for the comment. We have modified the text (line 53) to indicate the main advantage in the medical field; the following statements were added: “In terms of medical application, the processing of granules offers a huge advantage since no further filament making is necessary. Thus, no further manufacturing step has to be certified for medical purposes.”

And in line 66, the following was added to compare APF with FFF: “One significant advantage of the APF systems compared to FFF is the higher density that can be achieved in the fabricated. This higher density can lead to better mechanical performance and a smoother surface [2]”

Finally, the conclusion (line 342) now reads: “The research also shows the versatility of the APF process to use thermoplastic materials that have not been specifically tailored for an additive manufacturing process in their standard granular form.”

Conclusion:
In view of the above, the manuscript is not recommended for publication in its present form. The authors should modify the manuscript and address the above comments. In my opinion, a revised version, including above-mentioned points, may be considered for publication after reconsideration.

We appreciate  you valuable input.

Round 2

Reviewer 4 Report

Please use the following abstract for the second paper:

Abstract

The aim of this paper is to introduce tunable continuous-stable metamaterials with reversible thermo-mechanical memory operations by four-dimensional (4D) printing technology. They are developed based on an understanding on glassy-rubbery behaviors of shape memory polymers and hot/cold programming derived from experiments and theory. Fused decomposition modeling as a well-known 3D printing technology is implemented to fabricate mechanical metamaterials. They are experimentally tested revealing elastic-plastic and hyper-elastic behaviors in low and high temperatures at a large deformation range. A computational design tool is developed by implementing a 3D phenomenological constitutive model coupled with a geometrically nonlinear finite element method. Governing equations are then solved by an elastic-predictor plastic-corrector return map procedure along with the Newton-Raphson and Riks techniques to trace nonlinear equilibrium path. A tunable reversible mechanical metamaterial unit with bi-stable memory operations is printed and tested experimentally and numerically. By a combination of cold and hot programming, the unit shows potential applications in mimicking electronic memory devices like tactile displays and designing surface adaptive structures. Another design of the unit shows potentials to serve in designing self-deployable bio-medical stents. Experiments are also conducted to demonstrate potential applications of cold programming for introducing recoverable rolling-up chiral metamaterials and load-resistance supportive auxetics.

Author Response

1. Please use the following abstract for the second paper:

--> We thank the reviewer for the abstract provided; the abstract provided is for a paper that deals with the simulation of the 4D printing process, which is an important field of research but is not directly related to the current manuscript. We disagreed on adding a section on simulations of the 3D or 4D printing processes because it will distract from the main focus of the current experimental investigation.

Therefore, we will not cite the suggested paper and we will not modify the latest version of the manuscript, if there are no other comments.